# An intermittent detachment faulting system with a large sulfide deposit revealed by multi-scale magnetic surveys

Tao Wu [1], Maurice A. Tivey [2], Chunhui Tao [1,3 ✉], Jinhui Zhang [1], Fei Zhou [4] & Yunlong Liu [1]

Magmatic and tectonic processes can contribute to discontinuous crustal accretion and play an important role in hydrothermal circulation at ultraslow-spreading ridges, however, it is difficult to accurately describe the processes without an age framework to constrain crustal evolution. Here we report on a multi-scale magnetic survey that provides constraints on the fine-scale evolution of a detachment faulting system that hosts hydrothermal activity at 49.7°E on the Southwest Indian Ridge. Reconstruction of the multi-stage detachment faulting history shows a previous episode of detachment faulting took place 0.76~1.48 My BP, while the present fault has been active for the past ~0.33 My and is just in the prime of life. This fault sustains hydrothermal circulation that has the potential for developing a large sulfide deposit. High resolution multiscale magnetics allows us to constrain the relative balance between periods of detachment faulting and magmatism to better describe accretionary processes on an ultraslow spreading ridge.

[1] Key Laboratory of Submarine Geosciences, MNR, Second Institute of Oceanography, MNR, Hangzhou 310012, China. [2] Department of Geology and Geophysics, Woods Hole Oceanographic Institution, Woods Hole, MA 02543, USA. [3] School of Oceanography, Shanghai Jiao Tong University, Shanghai 200240, China. [4] Institute de Physique du Globe, CNRS UMR, 7154 Paris, France. ✉email: taochunhuimail@163.com

Accretion at mid-ocean ridges (MOR) has been documented to occur either in a symmetrical or asymmetrical mode[1,2]. Symmetrical accretion is most often dominated by magmatic processes, with high-angle normal faulting, and the formation of abyssal hills on both MOR flanks[3]. In contrast, asymmetric accretion typically shows an interplay between weaker magmatic and more prominent tectonic processes that is typical of both slow and ultraslow-spreading ridges[1,4,5]. In these spreading environments the development of detachment faults is the primary tectonic process that leads to thinning of the crustal section and exposure of lower crust and ultramafic mantle material on corrugated fault surfaces[6,7] that form oceanic core complexes (OCCs). These faults allow seawater to infiltrate deep into the footwall that can result in enhanced hydrothermal circulation and the deposition of polymetallic sulfide[8–10]. Repeated fault movement also allows for the permeability in the hanging wall to be reactivated periodically, which could control both the longevity of a hydrothermal system and its activity through time[11]. Thus, we would expect there to be a close interplay between hydrothermal activity and OCC formation and evolution.

Research on the structure and evolution of detachment faults has mainly focused on slow spreading environments of the Mid-Atlantic Ridge (MAR) with only a few examples from ultraslow-spreading environments such as the Southwest Indian Ridge (SWIR), and then only for areas that have little or no hydrothermal activity[12,13]. Few studies have investigated the detailed relationships between the evolution of detachment fault systems and their associated hydrothermal systems at ultraslow-spreading ridges mostly because it is difficult to accurately constrain the timing and because neovolcanic activity is broadly dispersed throughout the rift valley[14].

In this study, we attempt to provide quantitative constraints on the evolutionary history of a young detachment fault system and its associated hydrothermal systems. We analyze a series of high-resolution magnetic surveys (Fig. 1b)[15,16] from the Dragon Horn area on the ultraslow SWIR (49.7°E) where a deep-seated high-temperature hydrothermal circulation system has developed in close association with a major detachment faulting complex. Previous work has shown that the Dragon Horn detachment fault system penetrates to almost $13 \pm 2$ km depth below the seafloor and hydrothermal fluids circulate almost 6 km deeper than the Moho boundary[8]. It is unclear how long a hydrothermal system with such a deep circulation geometry and heat source could be sustained and whether it would result in a large sulfide deposit[17]. By combining near-bottom magnetic data and newly available sea-surface magnetic data with a large database of geological data (Fig. 1b), we attempt to understand and better constrain the geometry and evolution of an OCC and its associated hydrothermal circulation system. Our results show that high-resolution multiscale magnetic results at the Dragon Horn segment show that the older detachment fault (DF1) appears to have lasted approx. 0.72 Myr, and the present detachment fault (DF2) is likely only halfway through its evolution. The timeframe provides support for the hypothesis that hydrothermal circulation is mining heat over an extended period of time to support the development of a large sulfide deposit on this ultraslow-spreading ridge and allows us to better describe accretionary processes that are interspersed with significant periods of accretionary hiatus and tectonic extension.

## Results

**Sea-surface magnetics and seafloor spreading framework**. The three sea-surface magnetic profiles (L1–L3; Fig. 2a) have been reduced-to-the-pole (RTP) and show the polarity structure of the ridge segment in a regional sense. The positive normal polarity Brunhes magnetic anomaly is found over the axial volcanic ridge (AVR) and rift valley, while a reverse polarity magnetic anomaly is generally found over the rift valley walls of both the north ridge flank and the southern flank that encompasses the OCC dominated region. Through forward modeling, we identify normal polarity Chrons C1n and C2An and the intervening reverse Chrons C1r/C2r on both MOR flanks. We note that Chron C2n is only poorly resolved in the sea-surface profiles (Fig. 2). The boundaries of C1n and C2An are consistent with previous results[18,19], and allow us to calculate the average interval half-spreading rates for each ridge flank. Half-spreading rates for profiles L1, L2, and L3 (see Fig. 2c–e) are calculated from the middle of Chron C2An.1n (2.81 Myr) to the peak of the axial Brunhes anomaly (C1n, 0 Myr), which is assumed to be the time-averaged axis of spreading[20]. To the north, half-spreading rates show small differences between the profiles ranging from 7.48 to 6.82 to 7.01 km Myr$^{-1}$ (mean $7.10 \pm 0.34$ km Myr$^{-1}$) for L1 through L3, respectively. To the south, however, there are more obvious changes in half-spreading rates varying from 7.08 to 7.97 to 8.43 km Myr$^{-1}$ for profiles L1 through L3, respectively (mean $7.83 \pm 0.69$ km Myr$^{-1}$). The total mean opening rate is 14.84 km Myr$^{-1}$ for present day to Chron 2A, which is slightly faster than 13.89 km Myr$^{-1}$ reported by ref. [18] for the period from Brunhes to Chron 3A. While all profiles show some asymmetry, profile L3 shows the greatest asymmetry in half-spreading rate with ~20% asymmetry relative to the north. In fact, the Brunhes polarity appears to extend to the south over the OCC2 on this profile, which we interpret as normal polarity crust that has been exhumed on the detachment fault and translated to the south. By comparing profile L3 with the undisturbed L1 profile, (Fig. 2e), we find about 3.8 km of additional extension on the detachment fault of profile L3.

**Detailed observations of the OCC2 detachment fault**. The high-resolution bathymetry (Supplementary Fig. 1) provides a good starting basis for understanding the evolution of the Dragon Horn detachment fault system, which displays features that include: a mullioned structure on the OCC2 fault surface, a clear breakaway trace (B2), and the distinct definition of its termination (T2). Sea-surface magnetic profiles (Fig. 2) show that, in general, the south flank of the rift valley wall is within the reverse Matuyama polarity chron, with the exception of the Dragon Horn OCC2 itself, which, as noted earlier, has a small positive anomaly over it. The high-resolution near-bottom AUV magnetic RTP anomaly data (Fig. 3a) provide further details on this magnetic character with negative magnetic anomalies on the lower slopes of the OCC2 and a slightly more positive magnetic anomaly over the top of a stranded hanging wall block on the fault surface and a stronger positive anomaly just south of the breakaway B2 summit. These positive anomalies in the near-bottom data (Fig. 3), when continued upward to the sea-surface, appear to produce the small positive anomaly on the side of the Brunhes anomaly of the sea-surface profile L3 (Fig. 2). A zone of higher magnetic anomaly is also located at the western edge of the detachment fault on an apparent hanging wall block that hosts the LQ-1 vent field. Further off-axis, a zone of low magnetic anomaly is found associated with the termination of the older OCC (OCC1) in the southern part of the survey area (Fig. 3a).

A 3-D focused magnetization inversion (see "Methods") provides another view of the crustal magnetization structure and how it relates to the detachment fault and OCC. Depth slices from the focused inversion (Fig. 3c) show a weak magnetization over most of the OCC fault surface, a zone of positive magnetization associated with the top of the stranded block on

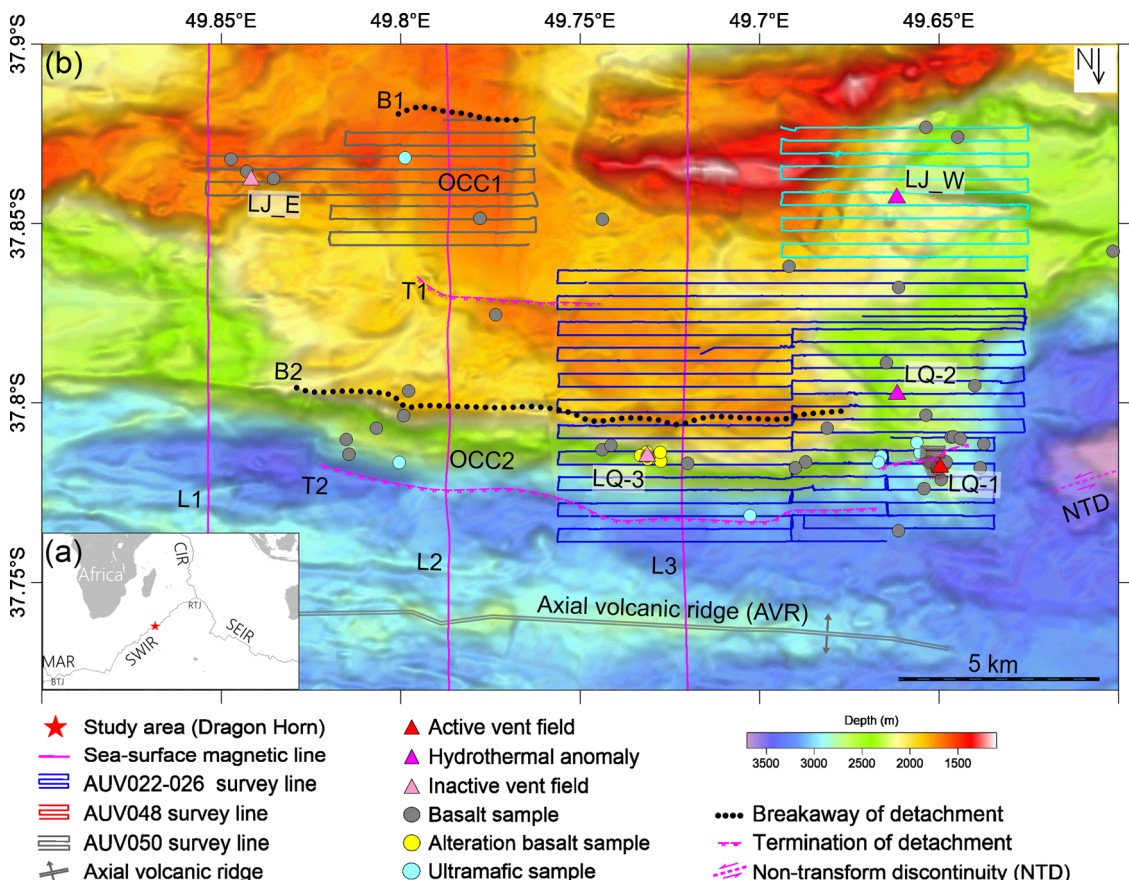

**Fig. 1 Bathymetry and geological setting of the Dragon Horn area segment 28 on the SouthWest Indian Ridge (SWIR). a** Regional location of the Dragon Horn area on the SWIR. **b** Summary bathymetry map with the distribution of hydrothermal fields (triangle markers) and geological tectonic features: black dots are the locus of the detachment fault breakaways, magenta hachured lines are the detachment fault terminations, OCC1 and OCC2 are oceanic core complexes exposed by fault slip, and NTD is the non-transform discontinuity. Sea-surface magnetic survey lines (L1, L2, L3) are shown by magenta lines. Autonomous underwater vehicle (AUV) tracklines are shown by blue, cyan, and gray lines.

the OCC2 surface, and a zone of stronger positive magnetization along the south side of the breakaway B2. The range of strong magnetization intensity of the stranded block does not appear to decrease with depth, which likely means that a relatively thick block along top edge (T2′) has detached from the initial breakaway at B2, where a zone of strong magnetization also exists (Fig. 3c). The talus zone at termination T2 at the base of OCC2 appears to show near-zero magnetization values likely indicating randomly oriented blocks compared to more coherently magnetized crust nearby.

In terms of the magnetic response of the hydrothermal systems on the seafloor crust, the inactive vent site LQ-3 located on the stranded block of OCC2 appears to have a muted magnetic signal. However, by extracting the raw east to west profile data (Fig. 3b) from across the vent site (double arrow line in Fig. 3a), we find a narrow but obvious magnetic anomaly low over the vent site about 200 m wide (Fig. 3b). Altered basalt was sampled from LQ-3, implying that focused hydrothermal alteration has only influenced the basaltic crust locally at this site. Further west, active vent site LQ-1 is located on the hanging wall of OCC2 (ref. [8]) but surveys (see Supplementary Note 1) do not show any clear magnetic low associated with LQ-1 (ref. [15]) and simply shows a relatively weak magnetization throughout the area that does not vary much with depth—at least to 238 m depth (Supplementary Fig. 2d). One possible interpretation of this result suggests that there has been pervasive alteration in this region such that no large magnetization contrasts are present in the upper part of the crust.

**Rock samples and physical properties.** Rock samples, collected with the Chinese TV grab, document the lithologies present on the OCC fault surfaces and adjacent terrain (Supplementary Table 1 and Fig. 4a, b). Basalt, gabbro, and peridotite samples are variously distributed across the OCC1 surface and show the influence of processes such as amphibolization, chloritization, and serpentinization. Specifically, as shown in Fig. 4a, b, ultramafic samples P1–P4 are all located on or near the OCC2 surface. P1 and P2 were collected from the LQ-1 vent site on the detachment fault surface, and the percentage of ultramafic rocks at P2 is higher but alteration is less pervasive than at P1, presumably because P2 is nearer to the termination T2 and thus from deeper in the crustal section and with less alteration time. P3 and P4 are located on the slip plane where the bend due to detachment tectonism and fracture caused by the decollement are documented in their deformation microstructure. Relatively fresh basalt and strongly altered peridotite are found at P5, which is probably the result of a talus accumulation at the base of the OCC slope from the collapsing section further upslope. The corresponding photomicrograph of P5 shows that the peridotite is strongly altered and contains magnetite. Samples from stations S1–S4, near the LQ-3 vent site, are basaltic rock that are heavily brecciated with different degrees of alteration. As noted earlier, venting at LQ-3 likely enabled hydrothermal fluid to alter the surrounding host rock in this area. The basalt is heavily brecciated and contains quartz and opaque iron minerals and/or is filled with chlorite. Banded alteration haloes are ubiquitous in the samples suggesting pervasive hydrothermal alteration.

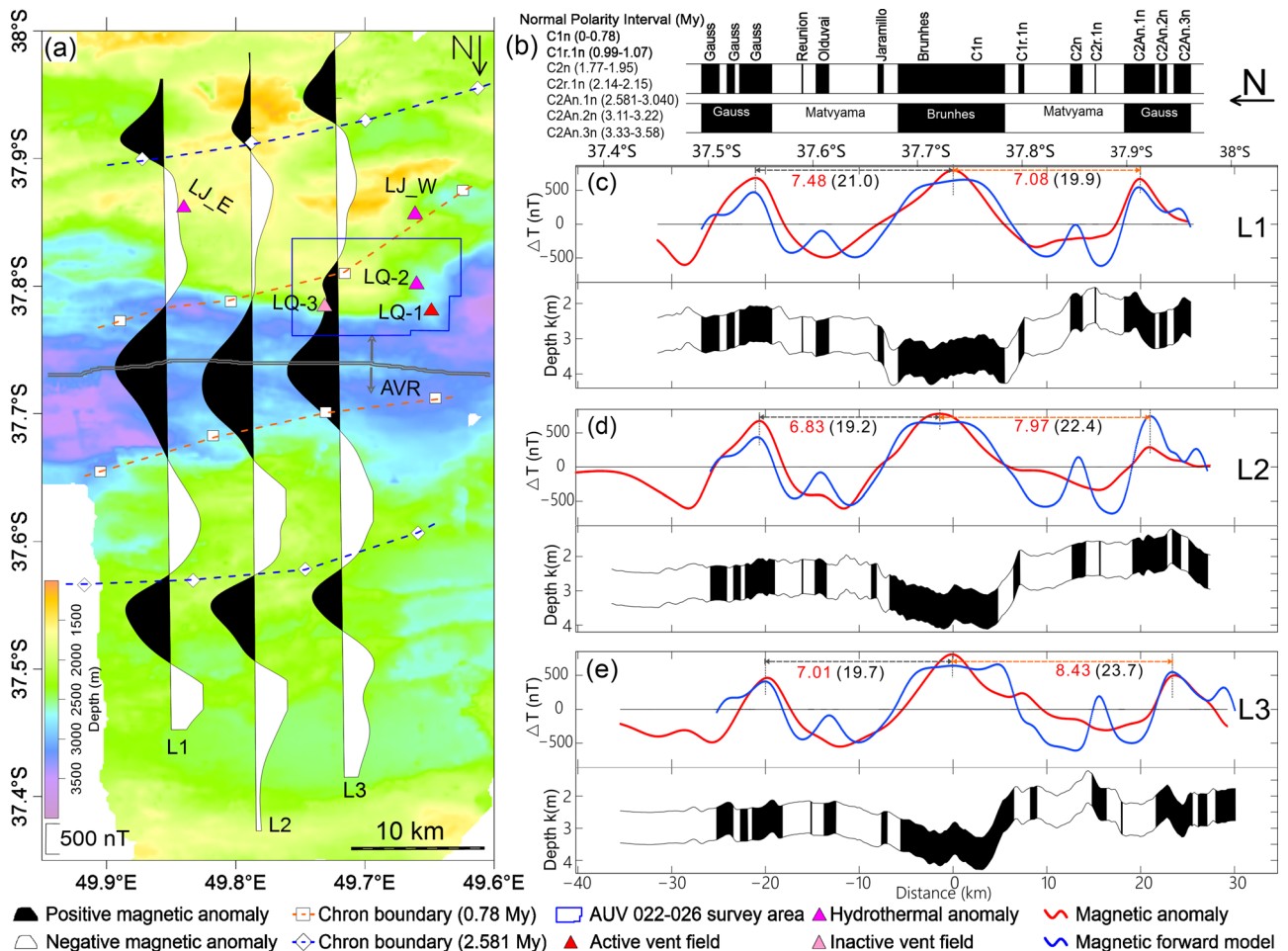

**Fig. 2 The regional magnetic polarity structure of the Dragon Horn ridge segment as documented by sea-surface magnetic survey profiles. a** Magnetic anomaly lineations. The boundaries of Brunhes (0.78 Myr) and Matuyama (2.581 Myr) obtained by ref. [19] are marked by pink and blue dashes respectively. **b** Geomagnetic polarity timescale[36]. **c–e** are comparisons between observed anomaly (red) and magnetic forward modeling results (blue) assuming a constant thickness source layer with 1 km. Where the figures in (**c–e**) noted by red color are half-spreading rates (in km Myr$^{-1}$) from the middle C2An.1n Chron to present-day and black numbers are the corresponding distances (in km).

**Forward modeling of OCC profile**. To gain more detailed insight into the evolution of the OCC2 region we forward modeled the observed near-bottom magnetic anomaly by assuming typical values for the magnetization of the various lithologies based on the recovered samples and to more realistically account for changes in the direction of the magnetic vector as the detachment fault rotates and as magnetic polarity reversals occur as part of the MOR accretion process[21]. We selected profile PP′ from the near-bottom data (see Figs. 3a and 4a) and forward modeled the magnetic field to match the survey results by building a 2-D structural magnetic model using the method of Luo and Yao[22] (Fig. 5a, b).

In modeling the magnetic anomaly signal along the profile, we targeted the following parameters to match the observed anomaly: (i) a magnetic contrast at the base and termination of the OCC between the exposed gabbro and peridotite and the basalts of the neovolcanic zone based on measurements of recovered samples (see Supplementary Table 1), (ii) incorporation of the polarity structure from the sea-surface magnetic spreading history (Fig. 2e), (iii) addition of mass wasting bodies that appear to have affected the crust along the profile. After setting these parameters, we find that the forward result (gray line in Fig. 5a) fails to match the observed data at the south end of the profile, especially between 37.815°S–37.83°S. However, we find a much better match (magenta line in Fig. 5a) once we consider adding in a positive magnetic zone between chrons C1n and C2An approximately 9 km from the AVR. This positive zone could be

either be the short Jaramillo event (0.99–1.07 Myr) or possibly the older and slightly longer C2n chron (1.77–1.95 Myr). The C2n solution would imply very slow and then very fast spreading between the event and the Brunhes (1.76 km Myr$^{-1}$) and the event and the C2A chron (15.47 km Myr$^{-1}$), respectively, which seems improbable. The Jaramillo solution results in a more even spreading rate between the event and the Brunhes and the event and the older C2A chron (7.6 and 8.26 km Myr$^{-1}$ respectively). Thus, we interpret this positive zone as most likely being Jaramillo in age. From the near-bottom magnetic anomaly map (Fig. 3a), we find that this positive zone extends along strike to the east, at least as far as 49.69°E (Fig. 3a). It is not visible in the sea-surface profile of L3 (Fig. 2e) at 49.72°E, although we see indication of an older anomaly that is likely chron C2n further south. Furthermore, using the distance between the Brunhes boundary and the present axis of spreading of 7.1 km, we estimate that the spreading half-rate that encompasses the OCC2 fault is 9.1 km Myr$^{-1}$, which is much faster than the subsequent accretionary spreading of 7.6 km Myr$^{-1}$ from Brunhes to Jaramillo where the distance is ~1.9 km according to Fig. 6. The detachment fault slip thus may be accommodating more to the overall asymmetrical spreading in this ridge segment. This local asymmetry in spreading rate reaches up to 30% faster relative to the normal flank spreading rate, which is greater than any asymmetry seen at the MAR[23].

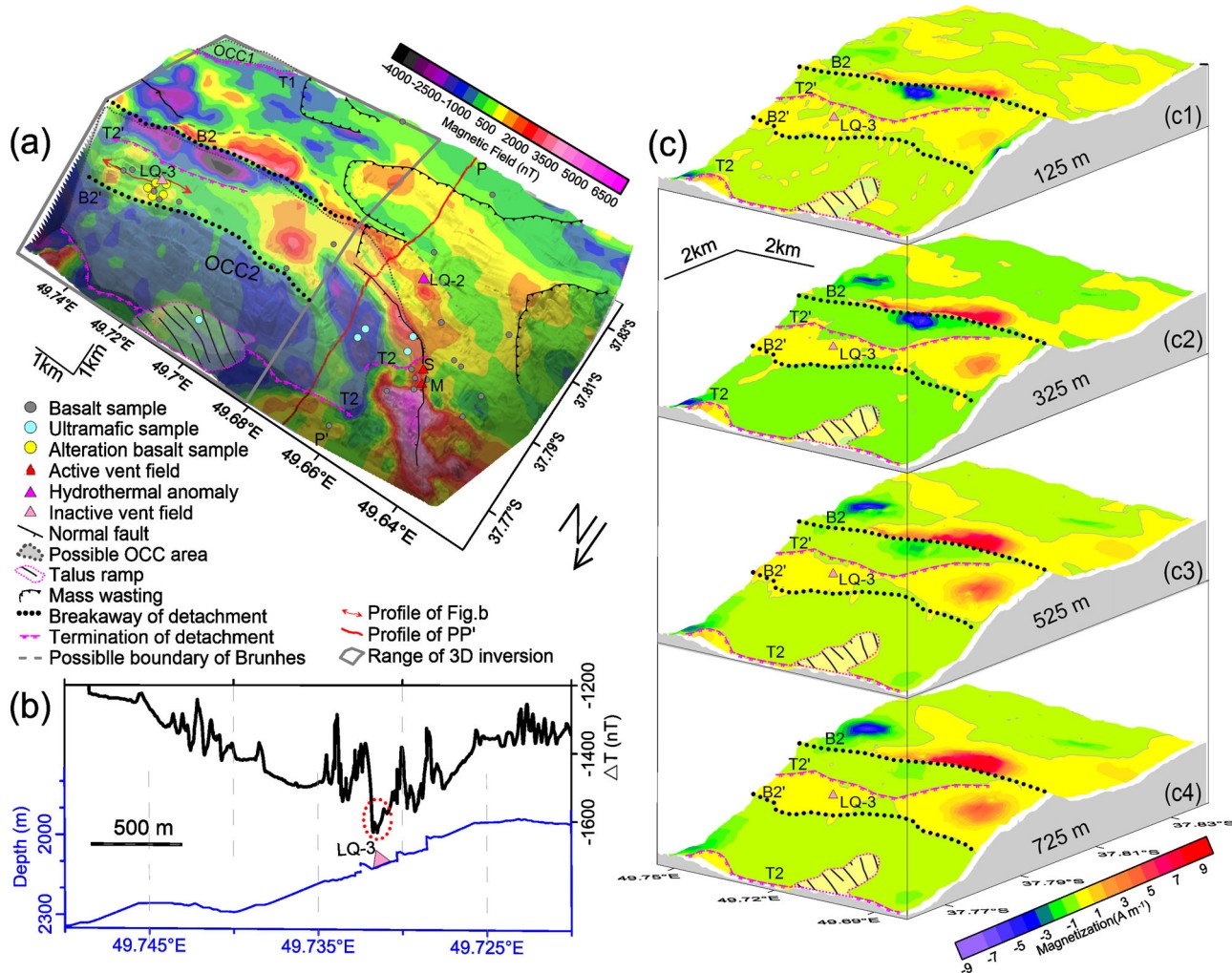

**Fig. 3 Near-bottom magnetic anomalies and results of magnetization inversion of the Dragon Horn OCC area. a** Near-bottom magnetic anomaly. **b** magnetic profile of a survey line which crosses vent field LQ-3 (marked by red double arrow line in **a**). **c** Focused 3-D magnetization inversion structure of the OCC2 system with **c**1, **c**2, **c**3, and **c**4 as the magnetization inversion results at depths 125, 325, 525, and 725 m, respectively. The geological body was divided into series of cells with 200 m (length) × 100 m (width) × 25 m (thickness) while performing magnetic inversion. We assume that the magnetization direction is parallel to the geocentric axial dipole (GAD) with the inclination and declination of −62.3° and −42.0°, respectively.

**Evolution of the Dragon Horn detachment fault**. With the detailed morphological mapping and magnetic constraints on spreading rate we can now attempt to reconstruct the evolutionary history for this portion of the ridge flank along magnetic profiles L2 and L3. For this reconstruction we assume that OCC formation initiates at the rift valley wall, typically located 3–6 km distant from the AVR accretionary axis (Fig. 6) and that OCC2 is still actively slipping as indicated by the seismic activity[24].

Starting with Profile L2, which has the best definition of both OCC1 and OCC2, we begin at the present day and restore the section sequentially through time. For OCC2 we restore the fault slip using the estimated half-spreading rate of 7.97 km Myr$^{-1}$ as calculated earlier from profile L2 (Fig. 2). The termination T2 of OCC2 is located approximately 5.9 km from the present-day axis of spreading. Given the lateral slip distance on OCC2 of 2.6 km results in an initiation age of 0.33 Myr before present (BP) for OCC2 (Fig. 6a). There is approximately 3.4 km between the breakaway B2 of OCC2 and the termination (T1) of the older OCC1. If we assume the older OCC1 formed at the same distance from the spreading axis as the present day OCC2 i.e. 5.9 km, this would translate into 0.43 Myr of spreading, suggesting that OCC1 (T1) stopped slipping approx. 0.76 Myr BP (i.e. 0.33 + 0.43 Myr).

Restoring the slip on OCC1 of 5.7 km at 7.97 km Myr$^{-1}$ gives a time duration of 0.72 Myr. Adding this time to the age at the end of slip on OCC1 gives an initiation time of 1.48 Myr BP for the formation of the OCC1 breakaway B2 (Fig. 6a).

Along profile L3, the older OCC1 is not well-defined; however, for the younger OCC2 we have noted that the Brunhes appears to be extended further here than the other profiles (Fig. 2), which is likely due to the formation of a stranded block (T2′–B2′, with ~500 m width) on the surface of OCC2. As noted earlier we have identified the Jaramillo event in the high-resolution near-bottom magnetic data (Fig. 5), which provides additional constraints on spreading half-rates. We estimate a half-spreading rate of 7.6 km Myr$^{-1}$ for the period between Jaramillo and the Brunhes chron, but a faster 9.1 km Myr$^{-1}$ for the Brunhes chron to the AVR, which encompasses the OCC2 fault. If we use the fast spreading half-rate of 9.1 km Myr$^{-1}$ for the slip on OCC2, again assuming it is currently active, we get a period of time of 0.31 Myr, slightly shorter than the estimate from profile L2 (Fig. 6b). If the fault initiated 0.31 Myr BP then there would have been 0.47 Myr of previous crustal spreading to create the Brunhes chron, which at the 9.1 km Myr$^{-1}$ rate would have resulted in 4.3 km of crustal accretion. Given that the fault termination T2 is

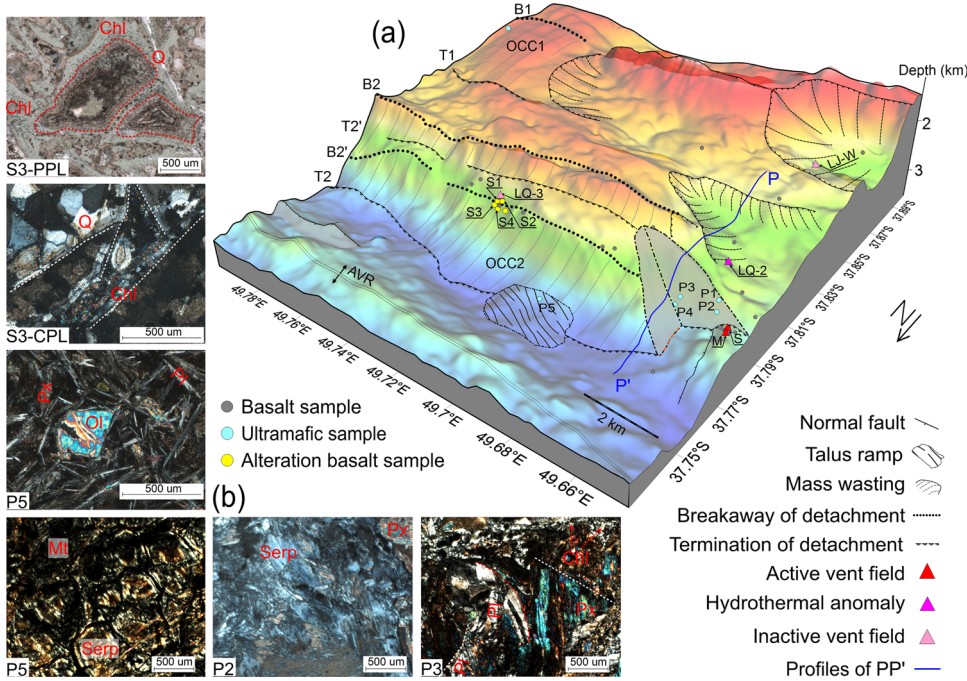

**Fig. 4 Interpretation of microphysiography of the Dragon Horn OCC2 system. a** The basic geologic and morphologic structure. detachment fault breakaways (B1, B2, B2') are shown by black dotted lines and terminations (T1, T2, T2') are shown by black hachured dashed lines. Thinner black hachured lines show areas of mass wasting and trajectories of collapse. Gray shaded areas shows the extent of the OCC2 fault surface near LQ-1. Lined area is a region of talus accumulated at the base of the OCC2 scarp. Triangle symbols are hydrothermal vent locations as described in the legend. Dots are sample locations with lithologies as described in Fig. 1 caption. Blue line is Profile PP' modeled in Fig. 5. **b** Photomicrograph and major minerals of representative rock samples which are observed in cross-polarized light (CPL), except the first (S3) with plane-polarized light (PPL). Sample S3, altered basalt filled with chlorite and a small amount of quartz (Q) in the ring banded structure and basalt detritus. Sample P5, peridotite is strongly altered and has formed serpentine and magnetite (Mt). Sample P2 has a network structure and mainly contains serpentine (Serp), residual peridotite, and pyroxenes (Px). Sample P3, bends and fractures are found and plagioclase (Pl), chlorite (Chl), and pyroxenes are present. Where the white dotted line is a fracture, red dotted curve line is a bend, and red enclosed line is a ring banded structure.

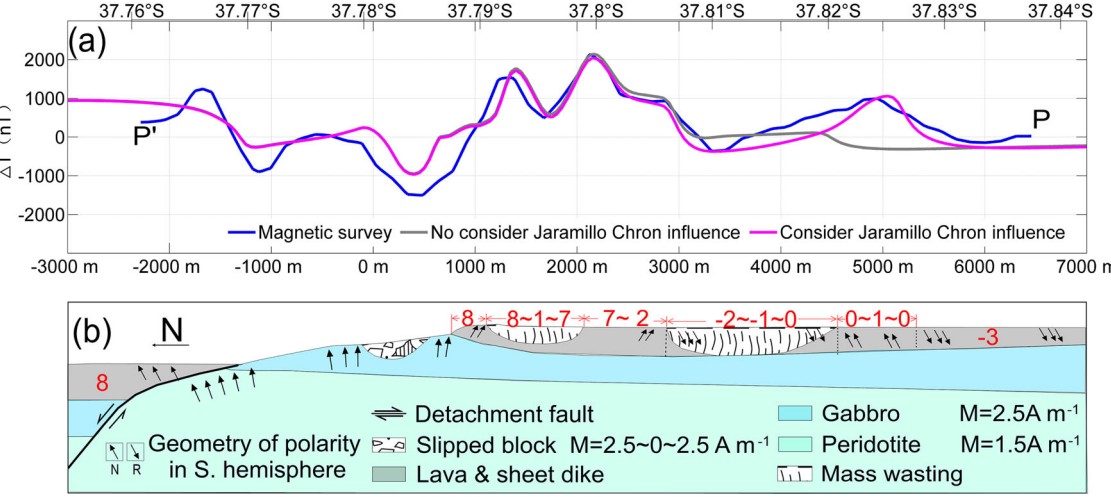

**Fig. 5 A comparison of forward magnetic modeling results based on 2-D magnetic models. a** Forward magnetic modeling results. **b** 2-D magnetic models with the observed near-bottom magnetic profile PP' (marked by bold blue line in **a**). We assumed that the effective magnetic layer thickness is 1 km, that the footwall rotated about 21.2° based on the present faulting surface slope angle, and that the rotation angle progressively increases with the distance from the termination. Red-colored numbers are magnetizations (in A m$^{-1}$) based on sample measurements (see Supplementary Table 1), where minus means reverse polarity. The mass wasting magnetization value of zero reduces the effective layer magnetization contribution. Geomagnetic reversals are modeled as vertical boundaries for simplicity and the GAD inclination (62.3°) is used except for the area changing with footwall rotation.

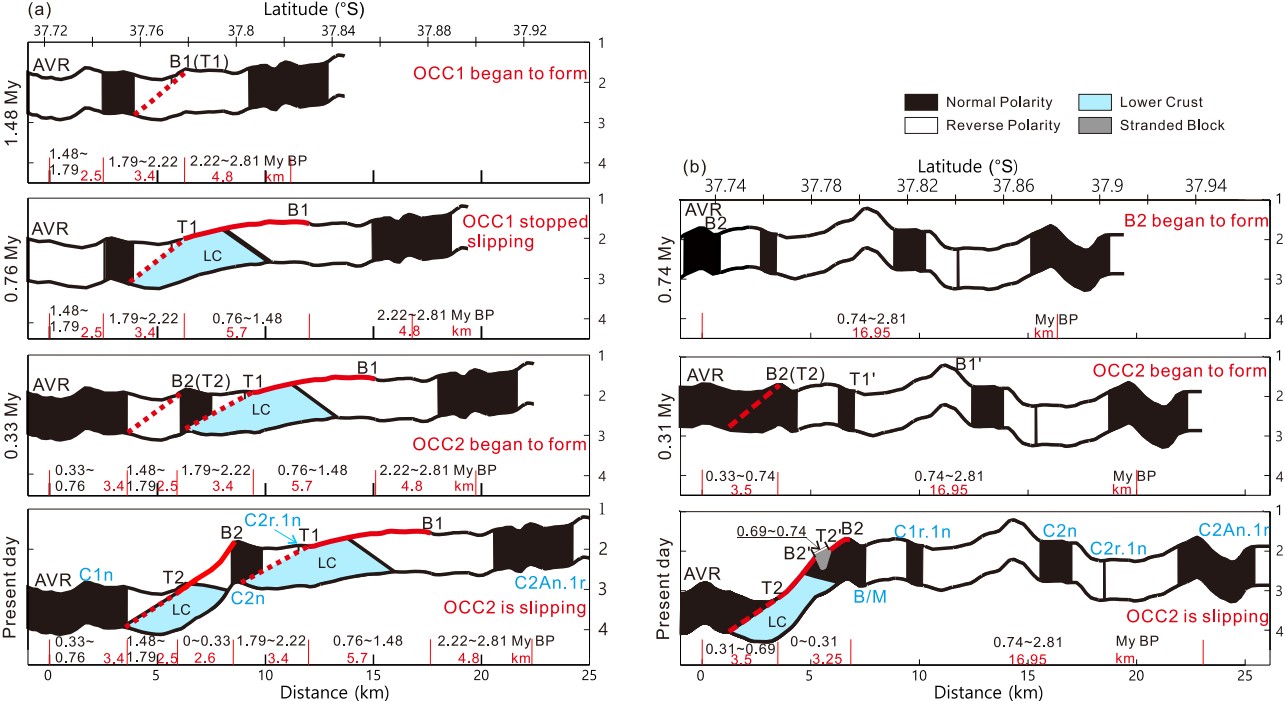

**Fig. 6 The evolutionary history of the detachment faulting system with hydrothermal activity based on profiles L2 and L3.** Only upper crustal section is shown with corresponding spreading rate along profiles and projected polarity zones. **a** Profile L2 evolution shown from 1.48 Myr BP to present and **b** Profile L3 evolution shown from 0.74 Myr BP to present. Also shown are the breakaway B1 and termination T1 of OCC1 and the breakaway B2 and termination T2 of OCC2. Red lines represent the OCC slip surfaces, other lines are accretionary terrain, and gray area represents the stranded block on OCC2. LC refers to lower crust exhumed by OCC1 and OCC2 and presumed to be less magnetic than the upper crustal extrusive section. Slip is assumed to be continuous along the OCC fault surfaces when they are formed until they terminate (i.e. no stop and start behavior). OCCs are assumed to nucleate off-axis approx. 3–6 km from the axis of spreading (NVZ). Topography is only approximate and may not reflect past topographic nature.

located 3.5 km from the axis (AVR) and adding the width of stranded block, then the fault clearly initiated within the normal polarity Brunhes chron, about 300 m from the Brunhes/ Matuyama boundary. This is consistent with the observation that the present-day fault breakaway B2 is ~300 m within the Brunhes chron (Fig. 5a). As noted in Fig. 1, Profile L3 does not have a clear expression of OCC1, which is present further west on Profile L2. Calculations from L2 suggest OCC1 initiated 1.48 Myr BP and terminated at 0.76 Myr BP, which would have obviated the accretion of Jaramillo-aged crust, if all the extension was taken up by slip on OCC1 along profile L2.

If we assume the periods when the OCCs were actively slipping were times of reduced magmatic supply, our evolutionary history for the Dragon Horn segment would imply that magmatic episodes of spreading occurred between 2.8 Myr and 1.48 Myr BP and again between 0.76 Myr and 0.33 Myr BP. Similarly, slip on the old detachment fault (OCC1) appears to have lasted approx. 0.72 Myr, which is within previous estimates of detachment fault slip in other SWIR segments 0.6–1.5 Myr[25]. The young detachment fault of OCC2 has been active for ~0.33 Myr and may continue for another ~0.4 Myr if the evolution time of DF1 is considered representative of a complete cycle.

## Discussion
With this framework for the OCC evolution and the timing constraints based on the magnetic data, we can now put the various hydrothermal systems present at Dragon Horn in a better context with respect to the history of accretion and tectonic activity and thereby reveal the control and interplay of processes responsible for the long-term evolution of hydrothermal vent and sulfide deposit formation. We believe that the most recent OCC2 at Dragon Horn formed in two main steps (Fig. 7). The original

detachment fault DF2 formed at the breakaway B2 (0.31–0.33 Myr BP) and stopped at termination T2′. Sometime afterwards, the main slip on the detachment fault stepped further to the south and initiated a second breakaway B2′ on a slip plane that stranded a block (T2′–B2′) on the surface of the scarp face (Figs. 4 and 5). The stranded block correlates with a modestly high magnetic anomaly along the middle of the detachment surface (Fig. 3). This block used to form part of the hanging wall but is now a stranded footwall block. The upper portion of the OCC2 surface, above T2′, is the fossil footwall, while the lower section of the OCC2 slip surface B2′ to T2 is now the exposed active fault (Fig. 7).

The inactive LQ-3 hydrothermal vent field, located on top of this stranded block on the mid-slope bench between B2′ and T2′, was likely active when the detachment fault initiated at B2 at 0.33 Myr BP, but likely became inactive when the new fault formed at B2′ and the block became stranded on DF2 and separated from its fluid circulation system and heat source. If we assume continuous slip on the OCC2 and partition time between the old and present-day slip surfaces simply based on the lateral distance, then we can estimate when LQ-3 might have stopped venting. The inactive fault surface (B2–T2′) is ~0.4 km wide. Using the interval half-spreading rate of 9.1 km Myr$^{-1}$ this would translate into the current slip occurring for ~46 kyr after which the main OCC detachment jumped from its previous location to its present location and formed the stranded block. We speculate that the LQ-3 vent site was actively venting up to 46 kyr prior to the faulting event that formed the stranded block and since that time the vent site has become inactive.

At the western edge of OCC2, the stranded block discussed above causes a lateral offset in the trace of the termination T2 (Fig. 4a). This offset in the trace of the termination means that the

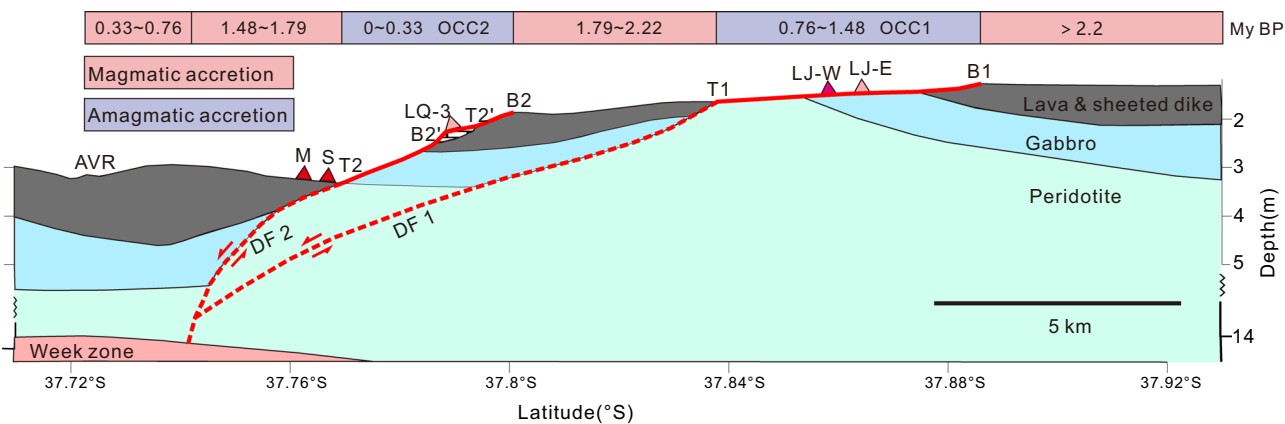

**Fig. 7 Intermittent detachment faulting and episodic magmatic accretion.** Present-day cross-section of profile L2 showing detachment fault geometry (red dashed lines) as determined by seismic data[24] and the surface expression of the faults (red solid lines) along with the inferred crustal units. The location of vent sites have been projected onto the cross-section and the LQ-3 vent site sits on the stranded block on the DF2 fault surface.

active fault surface of OCC2 is revealed upslope as a disconnected portion of the termination T2 at this western edge. Just down slope from termination T2 is the hanging block that hosts the LQ-1 vent field (Fig. 4a). The LQ-1 vent field includes the M and S zones. Human occupied vehicle (HOV) *Jiaolong* investigations suggest that the M zone is probably younger than the S zone as there are more inactive vents compared with the M zone, and the temperature of active vents in S are generally lower than those in M zone (Supplementary Fig. 3). The distribution of vent zones suggests a progressive migration of activity along a hydrothermal channel or fault zone[26] from the S zone to the M zone.

We speculate that the currently active LQ-1 vent field may have formed at the same time as the LQ-3 vent site but became inactive after the LQ-3 stranded block formed, but has subsequently resumed activity. This speculation is supported by detailed mineralogical patterns found in the chimneys and dating of the sulfides. A relict chimney collected from the M zone shows abundant medium-grained chalcopyrite as granular aggregates around the inner channels and fine- to medium-grained pyrite and sphalerite intergrown with minor chalcopyrite around the outer part of the chimney[27]. This indicates that the chimney had a low-temperature environment early in its formation as an outer pyrite-rich layer formed followed by a period of high temperature that formed the inner chalcopyrite-rich lining. We speculate that hydrothermal venting at LQ-1, although located on a hanging wall block, maybe linked to this stranded block formation. We suggest that the faulting activity forming the stranded block may have led to a rejuvenation of the thermal pathways for fluid flow and enhanced fluid discharge through the LQ-1 hanging wall block. Furthermore, as discussed earlier, if we assume the DF2 system may continue to be active and slip for another 0.4 Myr, this would allow hydrothermal circulation to tap heat over an extended period and allow LQ-1 to continue to grow into a large sulfide deposit.

As noted, additional hydrothermal activity appears to have occurred continuously during the evolution of the Dragon Horn detachment fault systems, We infer that the inactive LJ-E vent field may have been active sometime during 0.76–1.48 Myr BP when DF1 was active. The LJ-W hydrothermal anomaly was investigated[28] but no vent has been detected to date. We suggest that because heat is mainly being mined by the hydrothermal system related to the younger DF2 system, only a small amount of hydrothermal activity is focused through the path of the older DF1 system to the LJ-W site.

In summary, multiscale magnetic surveys are a useful approach for constructing a framework to accurately describe the timing of

magmatic and tectonic processes involved in the crustal accretion at ultraslow-spreading ridges. Specifically, high-resolution multi-scale magnetics allows us to constrain the relative balance between periods of detachment faulting and magmatism to better describe accretionary processes on an ultraslow-spreading ridge and the impact on hydrothermal processes. We find that detachment faults forming the OCCs appear to have a duration of ~0.72 Myr interspersed with shorter periods of accretion. This magmatic episodicity linked with the record of fault initiation and slip infers that the accretionary record has significant hiatuses with respect to distance from the AVR. As shown in Fig. 7, the age of crust between OCC1 and OCC2, i.e., B2 and T1 is 1.79 to 2.22 Myr, while the crust between the AVR and OCC2 (termination T2) is highly disrupted with ages of 0–0.33 (north of AVR with magmatic accretion), 0.33–0.76 and 1.48–1.79 Myr. Reference [14] has proposed a similar viewpoint according to U-series eruption ages of volcanic rocks collected from SWIR (11°–15°E). Our study shows that we can constrain these processes based on a detailed magnetic framework for detachment faulting systems on an ultraslow-spreading ridge. Finally, we find hydrothermal activity is intimately tied to the evolution of detachment faults and that vent systems could last for 0.72 Myr while others may be cut off from their heat sources through jumping fault systems and become inactive.

## Methods

**Sea-surface magnetic survey**. Sea-surface magnetic mapping during recent Chinese cruises collected additional profiles over the Dragon Horn ridge segment[29]. We processed these magnetic profiles by removing the International Geomagnetic Reference Field (IGRF)[30] and phase-shifting the anomaly profiles using a RTP transformation assuming a geocentric axial dipole (GAD) for the area (i.e. inclination of −62.3° and a declination of −42°) (Fig. 2). The easternmost survey line L1 appears to be unaffected by the Dragon Horn detachment fault system and so we can use it as a reference anomaly sequence for the area (Fig. 3). We carried out 2-D forward modeling to identify the magnetic anomalies using the geomagnetic polarity timescale[1] (Fig. 3). We assumed a constant 1000-m-thick magnetic layer with the upper surface defined by the bathymetry with a 9 A m$^{-1}$ magnetization for the Brunhes period (0–0.78 Myr) and a uniform ±6 A m$^{-1}$ magnetization for the off-axis crust. The effect of sloped polarity boundaries on magnetic anomaly amplitude has been ignored in the modeling.

**Near-bottom magnetic survey**. The first near-bottom magnetic data of the Dragon Horn area were collected by AUV *ABE* in 2007 (ref. [15]). Several Chinese research cruises visited the area more recently and have collected additional near-bottom magnetic data as part of a broader effort to document the nature and distribution of OCCs and hydrothermal systems of this region[31]. Between 2015 and 2018, seven AUV *Qianlong II* dives (AUV022–026, AUV048, and AUV050) were conducted in the Dragon Horn area during cruises DY40 and DY49 on board R/V *Xiangyanghong* 10 (Fig. 1b). Magnetic data were collected with 1 Hz sampling frequency, using a tri-axial fluxgate magnetometer installed on the stern of the

AUV *Qianlong II*[16]. In general, the AUV operated at approximately 100 m above the seafloor with a nominal track spacing of 400 m and an average transit speed of 1 m s$^{-1}$. High-resolution bathymetry and hydrothermal plume mapping data were collected concurrently during these surveys. Rock samples were obtained separately by the Chinese TV grab at locations marked in Fig. 1b.

AUV048 traversed over the LJ-W field and AUV050 made just one crossing of the LJ-E vent field and both were confined to traversing over the oldest terrain of the study area (OCC1). Therefore, we have primarily analyzed the near-bottom magnetic data over the youngest part of the terrain that includes OCC2 and its associated vent fields using five of seven AUV dives (AUV022–026). In addition to the magnetic analysis, we also measured the physical, geochemical, and microstructural properties of rock samples collected by the Chinese TV grab to help in the interpretation of the near-bottom magnetic anomalies. The study area includes all of the detected hydrothermal sites found by the HOV *Jiaolong*, the area surveyed by three of the *ABE* dives (ABE200–202) plus the multi-stage detachment OCC2 breakaway and termination region on the south flank of the Dragon Horn segment (Fig. 2). We reanalyzed the magnetic data from *ABE* dive 201 because it was surveyed at 50 m altitude and at a track spacing of 30 m, and thus provides a higher resolution dataset than the AUV *Qianlong II* surveys and can now be placed in better context with the broader mapping results.

Analysis of the AUV *Qianlong II* magnetic data included calibration and correction of the data for the magnetic effects of the vehicle and this was accomplished by spinning the AUV at the beginning of the dive once it had reached its operating depth. A five-factor trigonometric function method is used to calculate the calibration coefficients that are then applied to the measured magnetic data[32]. In order to account for any diurnal variations during the near-bottom magnetic surveys, a seafloor magnetometer was moored at a stationary location as a temporary geomagnetic station[16]. These diurnal variations were removed from the measured magnetic field followed by correction for the IGRF. We then applied filtering with a 7-s moving average filter (about 7 m averaging) to smooth the resultant magnetic anomaly data[26]. These AUV data were gridded with a minimum curvature method and RTP by assuming that the magnetization direction is parallel to the GAD for the area, as noted earlier (Fig. 3a). The influence of the fluctuating depth of the AUV survey lines was ignored for this RTP transformation. However, terrain undulation was considered in the further post processing inversion steps.

We used a 3-D focused inversion method to invert for a 3-D magnetization distribution with depth[33] (Fig. 3c). This inversion approach uses a regularized and conjugate gradient optimization technique to minimize the Tikhonov parametric function with a minimum support functional stabilizer[34] to obtain an optimized solution which is as geological realistic as possible. A depth-weighting function is employed to counteract the natural decay of the potential field with distance and to avoid singularities[35]. A terrain-weighting matrix was added to overcome the effect of undulating terrain on the inversion results. The study area was divided into a series of layers parallel to the terrain and then each layer was divided into a series of meshes. The magnetization of each layer was obtained through inversion and then these slices are displayed for their corresponding depth layer. For these inversions, we assumed GAD directions for the geomagnetic field and magnetization.

## Data availability
Sea-surface magnetic profiles, near-bottom magnetic grids, altitude, and bathymetry can be requested from the corresponding author C.T. (taochunhuimai@163.com). The lithologic characters and magnetization of samples in Dragon Horn collected by TV grab are supplied in the Supplementary Tables.

## Code availability
Computer codes and algorithms used in the current study are available upon request

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

## Acknowledgements

We thank Science Party, captains and crew of COMRA Cruise DY19th, DY40th, and DY49th, the 4,500 m Qianlong-II Autonomous Underwater Vehicle team and the submersible *Jiaolong* group. This work was supported by National Key R&D Program of China under contract no. 2018YFC0309901, National Natural Science Foundation of China (41906065), National Key R&D Program of China (2017YFC0306803) and COMRA Major Project under contract no. DY135-S1-01-06 and no. DY135-S1-01-01.

## Author contributions

T.W. carried out the investigation and the analysis, and wrote the manuscript. M.T. contributed to the analysis and the writing. C.T. conceived of the project and contributed the interpretation and writing. J.Z. participated in the investigation. F.Z. supplied the 3-D focused inversion method. Y.L. participated in the interpretation of the results. All authors participated in the review of this manuscript.

## Competing interests

The authors declare no competing interests.
