## [Peer Review File · Nature Communications]

REVIEWER COMMENTS

Reviewer #1 (Remarks to the Author):

The article entitled « An Intermittent detachment faulting system with a large sulfide deposit revealed by multi-scale magnetic surveys » has been written by Tao Wu, Maurice Tivey, Chunhui Tao, Jinhui Zhang, Fei Zhou and Yunlong Liu. It is proposed for publication in Nature Communication. The authors propose to use a multi-scale magnetic survey to constrain the evolution of a detachment faulting system that hosts hydrothermal activity.

The tectonics occurring on the vicinity of mid-ocean ridge is currently a very active field of research and detachment systems/Oceanic Core Complexes are clearly a top priority. Many questions remain to be answered and the way they develop over time is still poorly constrained. By providing a detailed reconstruction of their evolution, the authors therefore offer a very interesting approach and I have been totally convinced by their work.

I only have minor comments that, I think, are very useful for the reader:

Line 114: Which type of inversion have you used. This is not clearly stated and this is something important because depending on the type of inversion you use; a problem could arise with deep-sea data collected by an AUV that does not acquire the data on a horizontal plane. Have you used an inversion that takes these immersion variations into consideration? Have you upward-continued the data to a horizontal datum plane to suppress these immersion variations? In such case, do you have an estimation of the loss of the wavelength content of the signal?

Line 116: Something is not clear with the depth slices. How do you perform such slices? The depth of the dominating sources you see on a magnetic signal is correlated to the altitude of the measurement but if I understand properly, you do not have AUV dives at several altitudes above the seafloor. In such case, how can you deduce the horizontal variations of magnetization at different depths? Are you performing an inversion with virtual bathymetric grids located at fixed depths under the seafloor? If it's the case, then I would find this approach very questionable. This really lacks clarity and needs to be properly explained because at the moment, I find it very confusing as a reader and I cannot assess the precision of these results.

Line 174: Here the authors mention forward models. This is a typical approach that is often used for potential fields. Nevertheless, such models are very dangerous because an infinite number of models lead to satisfying results. As a consequence, I recommend the authors should clearly explain which hypotheses they have chosen (at the moment they remain quite vague) and the parameters they have used to compute the model. Once again, the impression I got is that we just have to believe what the authors have done (which I do not question because I find their work very good) but without clearly showing it. SO please, I encourage you to be more precise at explaining your forward model.

These are the comments I wanted to say. But besides that, this is a very good paper and I am totally convinced it should be published in Nature Communications after these minor adjustments have been completed.

Reviewer #2 (Remarks to the Author):

This paper describes geomagnetic data from a detachment fault on the ultraslow SW Indian Ridge. This area has already been introduced in Nature Communications in a paper by Tao et al. "Deep high-temperature hydrothermal circulation in a detachment faulting system on the ultra-slow spreading ridge"
The first paper describes mainly OBS and fluid chemical data, defining the geometry of multiple detachments and inferring deep circulation from geochemical data. The current submission is based on a combination of

seaborne magnetic data and AUV surveys. So a considerable amount of data is presented and it certainly should be published.

Asymmetric spreading is identified from the spacing of magnetic anomalies – this is normal in detachment mode oceanic spreading, but has probably not been previously demonstrated from a ultraslow ridge. I found the description and conclusions from the AUV studies combined with forward modelling quite hard to follow but this is not my field. Having established some ages for different detachment faults, there is a very detailed description of the evolution of faulting and hydrothermal activity in the area. For me, this did not bring out any new findings of global significance, although as a detailed integration of magnetic and geological data it is interesting.

To summarise, the area has already been described in Nature Comms and many of the more important results cherry-picked. This is a more detailed and rather parochial study where the conclusions do not go far beyond the specific area. I think it would be more suitable in a journal such as G-cubed, but even there the importance of the results needs to be brought out.

Andrew McCaig

Reviewer #3 (Remarks to the Author):

Key results:

This paper uses modelling of sea-surface and seafloor magnetic anomalies over an area of the SW Indian Ridge that is characterised by a complex and long-lived oceanic detachment fault system to quantify slip-rates on these plate-bounding faults and their evolution with time. The authors then infer that associated high-temperature hydrothermal systems were continuously active throughout the slip history of the two successive detachment faults under study, suggesting that these structures provide long-term pathways that allow hydrothermal fluids to mine heat at depth.

Validity:

The data presented in the paper are valid and of a high quality. The analysis is robust, although the discussion and presentation of the main results needs some work to highlight the most important new findings.

Significance:

There is much interest in the evolution of oceanic detachment faults/oceanic core complexes through time, following their discovery in the 1990's and the recognition that such systems reflect a fundamentally different detachment-mode of seafloor spreading than normal magmatic accretion. It's rare for a study to be able to provide temporal constraints on the displacement history of these important structures, especially at ultra-slow spreading rates. In this case, the authors then use their magnetic age constraints to make inferences about the longevity and history of associated hydrothermal systems, that are themselves poorly understood. The study therefore has the potential to be a highly significant case study, but it needs some extensive restructuring/rewriting to allow this significance to stand out.

Data and methodology:

The paper presents new and useful data on both sea-surface and seafloor magnetic anomalies. The methodology section gives a very clear account of the data acquisition, reduction and modelling techniques that have been employed.

Analytical approach:

The authors present a very thorough discussion of the stages in their analysis (although some of these details detract from the overall messaging). I would have liked to see some discussion about the validity of their assumption of a magnetic layer source layer with a constant thickness of 1 km that lies at the heart of the forward modelling – how does this assumption work over the oceanic core complexes in the study area that have detachment faults that have exhumed deeper crustal/mantle levels to the seafloor and that are likely to have displacements > 1km. Surely this would result in an uneven thickness of source layer,

especially as it is known that the evolution of these faults involves rotation of their footwalls.

Suggested improvements:

The paper is difficult to read effectively at the moment because of (a) excessive use of abbreviations throughout (e.g. LQ-1 to LQ-3, DF, LJ-E, LJ-W, T1, T2, T2', B2, B2', Vc1s, Vc1n, P1 to P4, S1 to S4) and (b) a general lack of clear "story-telling" resulting in the overall message being lost within the discussion of the subtleties of the analysis and interpretation. The authors could usefully shorten the text to improve the focus, and I suggest that they approach this by thinking about and listing the major points that they want to make in each section, before working on the text to emphasize these points. Some text could potentially also be usefully moved to supplementary information to avoid clutter.

Clarity and context:

This paper is well-written overall (spelling and grammar are fine), but the use of multiple abbreviations throughout detracts from the main messages that the authors are trying to get across (see suggested improvements).

References:

The paper adequately references the existing literature.

Our response to the reviewer's comments are as follows (we repeat their comments and show our responses in a blue color)

Reviewer #1 (Remarks to the Author):

The article entitled « An Intermittent detachment faulting system with a large sulfide deposit revealed by multi-scale magnetic surveys » has been written by Tao Wu, Maurice Tivey, Chunhui Tao, Jinhui Zhang, Fei Zhou and Yunlong Liu. It is proposed for publication in Nature Communication. The authors propose to use a multi-scale magnetic survey to constrain the evolution of a detachment faulting system that hosts hydrothermal activity.

The tectonics occurring on the vicinity of mid-ocean ridge is currently a very active field of research and detachment systems/Oceanic Core Complexes are clearly a top priority. Many questions remain to be answered and the way they develop over time is still poorly constrained. By providing a detailed reconstruction of their evolution, the authors therefore offer a very interesting approach and I have been totally convinced by their work.

I only have minor comments that, I think, are very useful for the reader:

Ans : We thank the reviewer for their kind remarks. Before addressing specific reviewer requests, we wish to point out that the methods we used in this study are stated in section on Methods located at the end of the main manuscript. We have added some more details but have tried to be concise in explaining our methods and forward models with references to readers to follow up on details. Now, we would like to respond to your specific requests as following

Line 114: Which type of inversion have you used. This is not clearly stated and this is something important because depending on the type of inversion you use; a problem could arise with deep-sea data collected by an AUV that does not acquire the data on a horizontal plane. Have you used an inversion that takes these immersion variations into consideration? Have you upward-continued the data to a horizontal datum plane to suppress these immersion variations? In such case, do you have an estimation of the loss of the wavelength content of the signal?

Ans: the inversion we used does take the effect of the terrain and depth of AUV (i.e. immersion) into consideration. We have added more details of the inversion approach in the last paragraph of Methods section. Also, please see our response to the next question for more details of the inversion approach.

Line 116: Something is not clear with the depth slices. How do you perform such slices? The depth of the dominating sources you see on a magnetic signal is correlated to the altitude of the measurement but if I understand properly, you do not have AUV dives at several altitudes above the seafloor. In such case, how can you deduce the horizontal variations of magnetization at different depths? Are you performing an inversion with virtual bathymetric grids located at fixed depths under the seafloor? If it's the case, then I would find this approach very questionable. This really lacks

clarity and needs to be properly explained because at the moment, I find it very confusing as a reader and I cannot assess the precision of these results.

Ans: As mentioned in the last paragraph of the Methods section, we used a 3-D focused inversion method to invert for a 3-D magnetization distribution with depth (see Zhou et al., 2018 for details of the approach). This inversion approach uses a regularized and conjugate gradient optimization technique to minimize the Tikhonov parametric function. Although this is an underdetermined problem (observation points are less than inverse magnetization model parameters) and the solution is classically non-unique, we use the minimum support functional stabilizer introduced by Portniaguine and Zhdanov (1999) to obtain an optimized solution which is as geologically realistic as possible. The intrinsic decay of magnetic field with distance was compensated by a depth-weighting functional. In addition, a terrain-weighting matrix was added to overcome the effect of undulating terrain on the inversion results. In particular, the study area was divided into series layers which parallel the terrain and then each layer was divided into a series of meshes. The magnetization of each layer was obtained through inversion and then these slices can be mapped to the corresponding depth layer inversion results. Similar methods have been used in magnetic inversions of hydrothermal vents studies such as Galley et al., 2020 and Caratori Tontini et al., 2012.

Galley, C. G., et al., Magnetic imaging of subseafloor hydrothermal fluid circulation pathways. *Science Advances*. **6(44)** (2020).

Zhou, F., et al., 3D Focused Inversion of Near-bottom Magnetic Data from Autonomous Underwater Vehicle in Rough Seas. *Ocean. Sci. J.* **53**, 405-412 (2018).

Portniaguine, O. & Zhdanov, M. S. Focusing geophysical inversion images. *Geophysics* **64(3)**:874–887 (1999).

Caratori-Tontini, F., et al., 3-D focused inversion of near-seafloor magnetic data with application to the Brothers volcano hydrothermal system, Southern Pacific Ocean, New Zealand. *Journal of Geophysical Research Solid Earth*. **117(B10)** (2012).

Line 174: Here the authors mention forward models. This is a typical approach that is often used for potential fields. Nevertheless, such models are very dangerous because an infinite number of models lead to satisfying results. As a consequence, I recommend the authors should clearly explain which hypotheses they have chosen (at the moment they remain quite vague) and the parameters they have used to compute the model. Once again, the impression I got is that we just have to believe what the authors have done (which I do not question because I find their work very good) but without clearly showing it. SO please, I encourage you to be more precise at explaining your forward model.

These are the comments I wanted to say. But besides that, this is a very good paper and I am totally convinced it should be published in Nature Communications after these minor adjustments have been completed.

Ans: We appreciate that forward models are non-unique and that we cannot claim to know the true source of the anomalies, however, the key piece of information from this part of the study is the

recognition of the Jaramillo anomaly in the profile, which helps us define part of the temporal framework. We have tried to state the assumptions made in the forward modeling more clearly in the main text (lines 192 to 201) and in figure caption (Fig. 5). We have used as much information as we have from geological observations (for mass wasting regions and fault slip) and the values for the crustal magnetization from paleomagnetic measurements of recovered rock samples. As we note in the text (line 191 marked-up version) we use the method of Luo (2007) for the forward modeling. This approach introduces the Euler equation into the theoretical expression of magnetic field of a cuboid source body and its gradient field without analytic singular points. For the 2-D models, the length of the cuboids (Vertical direction of the profile) were set to a long enough length so that they could be treated as 2-D bodies.

Reviewer #2 (Remarks to the Author):

This paper describes geomagnetic data from a detachment fault on the ultraslow SW Indian Ridge. This area has already been introduced in Nature Communications in a paper by Tao et al. “Deep high-temperature hydrothermal circulation in a detachment faulting system on the ultra-slow spreading ridge”

The first paper describes mainly OBS and fluid chemical data, defining the geometry of multiple detachments and inferring deep circulation from geochemical data. The current submission is based on a combination of seaborne magnetic data and AUV surveys. So a considerable amount of data is presented and it certainly should be published.

Asymmetric spreading is identified from the spacing of magnetic anomalies – this is normal in detachment mode oceanic spreading, but has probably not been previously demonstrated from a ultraslow ridge.

I found the description and conclusions from the AUV studies combined with forward modelling quite hard to follow but this is not my field. Having established some ages for different detachment faults, there is a very detailed description of the evolution of faulting and hydrothermal activity in the area. For me, this did not bring out any new findings of global significance, although as a detailed integration of magnetic and geological data it is interesting. To summarise, the area has already been described in Nature Comms and many of the more important results cherry-picked. This is a more detailed and rather parochial study where the conclusions do not go far beyond the specific area. I think it would be more suitable in a journal such as G-cubed, but even there the importance of the results needs to be brought out.

Andrew McCaig

Ans : We thank the reviewer for their remarks and for their specific suggestions noted in an annotated manuscript. We have replied and revised the manuscript according to the reviewer comments. We also noted in the previous annotated manuscript (PDF document, 2_reviewer_attachment_1_1612732174_convrt). Our first submitted version was somewhat difficult to read and follow, so we have re-structured the manuscript and made clearer points at each subsection. In addition, we would like to emphasize the broader significance of the work. The use of a multiscale magnetics approach to produce a timeframe for interpreting the history of the OCC formation and detachment fault slip is, we believe, a novel development. While the methods and application are to this particular site at Dragon Flag, we believe these data provide insight into the timescales for detachment fault evolution in general, and its links to hydrothermal vent activity and sulfide mineral deposition that is of broad interest. Some points we wish to emphasize are:

1. So far, few studies have investigated the detailed relationships between the evolution of detachment fault systems and their associated hydrothermal systems at an ultraslow spreading ridge, as it is difficult to accurately constrain the timing even when a comprehensive U-series dating program is attempted (Standish and Sims, 2010, NG). Our study uses multi-scale magnetics

to identify the precise location of polarity blocks and a short polarity chron that allows us to constrain the fine-scale evolution of detachment faults, crustal accretion, and the history and evolution of the associated hydrothermal areas. A short polarity chron, such as the Jaramillo, often only has few hundred meters width and is difficult to recognize in sea-surface magnetic data when the seafloor is several thousands of meters deep, however near-bottom surveys makes the identification of these short events more feasible. This approach is a new attempt and can be broadened to other ridges.

2. As stated in the Introduction, the previous Nature Comms article (Tao et al., 2020) has shown that the Dragon Horn detachment fault system penetrates to almost 13 ± 2 km depth below the seafloor and hydrothermal fluids circulate almost 6 km deeper than the Moho boundary. However, without a constraint on timing, it is still unclear how long a hydrothermal system with such a deep circulation geometry and heat source could be sustained and whether it would result in a large sulfide deposit. Our results address these questions more directly and provide an overall framework for the evolution of long term venting and sulfide mineral formation.

3. Finally, our results suggest that intermittent detachment faulting interspersed with episodic magmatic accretion will result in an accretionary record that has significant hiatuses with respect to distance from the AVR. As stated in the Discussion section, Standish and Sims (2010, Nature Geoscience) proposed a similar viewpoint based on U-series eruption ages of volcanic rocks collected from SWIR (11° – 15° E). They suggested that existing models may not accurately describe crustal accretion at ultraslow-spreading ridges. Our study shows that we can potentially constrain the timescale of these processes based on a detailed magnetic framework for detachment faulting systems on ultraslow-spreading ridges.

In short, we believe that our study could appeal to the interest of widest possible international geological and geophysical audiences.

Reviewer #3 (Remarks to the Author):

Key results:

This paper uses modelling of sea-surface and seafloor magnetic anomalies over an area of the SW Indian Ridge that is characterised by a complex and long-lived oceanic detachment fault system to quantify slip-rates on these plate-bounding faults and their evolution with time. The authors then infer that associated high-temperature hydrothermal systems were continuously active throughout the slip history of the two successive detachment faults under study, suggesting that these structures provide long-term pathways that allow hydrothermal fluids to mine heat at depth.

Validity:

The data presented in the paper are valid and of a high quality. The analysis is robust, although the discussion and presentation of the main results needs some work to highlight the most important new findings.

Significance:

There is much interest in the evolution of oceanic detachment faults/oceanic core complexes through time, following their discovery in the 1990's and the recognition that such systems reflect a fundamentally different detachment-mode of seafloor spreading than normal magmatic accretion. It's rare for a study to be able to provide temporal constraints on the displacement history of these important structures, especially at ultra-slow spreading rates. In this case, the authors then use their magnetic age constraints to make inferences about the longevity and history of associated hydrothermal systems, that are themselves poorly understood. The study therefore has the potential to be a highly significant case study, but it needs some extensive restructuring/rewriting to allow this significance to stand out.

Data and methodology:

The paper presents new and useful data on both sea-surface and seafloor magnetic anomalies. The methodology section gives a very clear account of the data acquisition, reduction and modelling techniques that have been employed.

Analytical approach:

The authors present a very thorough discussion of the stages in their analysis (although some of these details detract from the overall messaging). I would have liked to see some discussion about the validity of their assumption of a magnetic layer source layer with a constant thickness of 1 km that lies at the heart of the forward modelling – how does this assumption work over the oceanic core complexes in the study area that have detachment faults that have exhumed deeper crustal/mantle levels to the seafloor and that are likely to have displacements > 1km. Surely this would result in an uneven thickness of source layer, especially as it is known that the evolution of

these faults involves rotation of their footwalls.

Ans: We thank the reviewer for their positive comments on our work.

We agree that thickness of the magnetic source layer is an important consideration. While the reviewer focused on the forward modelling we would like to point out here that our focused inversion of the near bottom AUV data indeed considers the thickness and we show this through the depth slices. Furthermore, previous studies show that, once the depth of the magnetic source is ~5 times deeper than the survey altitude (Fujii et al. 2015), the corresponding magnetic anomaly is too weak for our survey and can be ignored as a contribution given the quadratic/ cubic attenuation with the distance between the platform and magnetic source.

The reviewer comments on the forward modeling are valid in that we have chosen not to vary the magnetic source layer thickness. The reasoning for this was that the goal of the forward modeling was primarily to map the location of polarity boundaries and investigate the effect of fault rotation on the magnetic vector. Clearly the magnetic source layer thickness is a factor but its effect is, to some extent, included in the variable magnetization, which could easily be mapped into thickness variations. We acknowledge the limitations of our approach but argue that the recognition of the Jaramillo chron is the key result along with the extent of the Brunhes chron, which we believe is a robust result regardless of whether the source layer varies in thickness or lateral changes in the magnitude of magnetization.

Fujii, M., et al., High-resolution magnetic signature of active hydrothermal systems in the back-arc spreading region of the southern Mariana Trough. *J. Geophys. Res. Solid Earth*. **120** (2015).

Suggested improvements:

The paper is difficult to read effectively at the moment because of (a) excessive use of abbreviations throughout (e.g. LQ-1 to LQ-3, DF, LJ-E, LJ-W, T1, T2, T2', B2, B2', Vc1s, Vc1n, P1 to P4, S1 to S4) and (b) a general lack of clear "story-telling" resulting in the overall message being lost within the discussion of the subtleties of the analysis and interpretation. The authors could usefully shorten the text to improve the focus, and I suggest that they approach this by thinking about and listing the major points that they want to make in each section, before working on the text to emphasize these points. Some text could potentially also be usefully moved to supplementary information to avoid clutter.

Ans: (a) We have reduced many of the abbreviations present in the text to make the manuscript more read-able. For example, we have deleted abbreviations Vc1s and Vc1n, and reduced the use of DF as an abbreviation of detachment fault. However, we have retained the use of DF1 and DF2

to distinguish the old and the new detachment fault. We still use T1, T2, T2', B2, B2' to represent the termini and breakaways of the old and the new detachment faults respectively, which lets the paper be read more effectively now. Certainly, LQ-1 to LQ-3 , LJ-E and LJ-W are used to distinguish the different vents and they are follow their names that have represented in previous literature. P1 to P4 and S1 to S4 are just characters to present the different samples and to convenient description.

(b) To provide our work with a better “story-telling” thread, we have tried to make clearer points within each subsection. We have deleted subheadings in the Discussion section and moved the text about the Evolution of the Dragon Horn detachment fault from the Discussion and placed it in the Results section. To improve clarity on the significance of our work, Figure 7 was separated from the previous Figure 6 and is used to describe how the timing of magmatic and tectonic processes involved in the crustal accretion at ultraslow-spreading ridges.

Clarity and context:

This paper is well-written overall (spelling and grammar are fine), but the use of multiple abbreviations throughout detracts from the main messages that the authors are trying to get across (see suggested improvements).

Noted as above

References:

The paper adequately references the existing literature.

REVIEWERS' COMMENTS

Reviewer #1 (Remarks to the Author):

The authors have carefully addressed my comments.

All the questions I was wondering about now have a proper answer. Consequently I think the article should be ready for publication at Nature Communications.

Reviewer #3 (Remarks to the Author):

The authors have done an excellent job of taking the various comments of the reviewers into account in this revised version. The paper is now much more impactful and readable, and I believe it is now suitable for publication in Nature Communications in its current form. I have one very tiny correction that could be dealt with at the proof stage - change "it's" in the abstract to "it is". Other than that, the paper is in excellent shape for publication!

Dear Reviewers:

We are glad to hear that our manuscript is on the way to be published. We thank for your approval.

As only Reviewer #3 had a tiny correction that is change "it's" in

the abstract to "it is", we have accepted the advice and have corrected.

In addition, we have completed all the requirements by editor, as following

- Have uploaded checklist describing our response to editorial requests.
- Have submitted the final version of our manuscript as a Word with all changes highlighted in the text.
- Have provided the complete author list in the manuscript file.
- Have uploaded production-quality versions of each figure as a separate file containing all panels.
- All updated checklists have been verified compliance with our research ethics and data reporting standards in PDF format.
- The final version of the Supplementary Information have been in one PDF file.
- Haven't submitted any Supplementary Movie, Audio, Data and Software.
- Each image has only been used one form and been given a scientific description of the image in the 'title' field.

At last, thank again for your hardworking and approval.

That's all. Best regards.

Sincerely,

Dr. Chunhui Tao, Corresponding Author

On the behalf of all co-authors

13th, July. 2021